# Acacia Gum Is Well Tolerated While Increasing Satiety and Lowering Peak Blood Glucose Response in Healthy Human Subjects

**DOI:** 10.3390/nu13020618

**Published:** 2021-02-14

**Authors:** Riley Larson, Courtney Nelson, Renee Korczak, Holly Willis, Jennifer Erickson, Qi Wang, Joanne Slavin

**Affiliations:** 1Food Science and Nutrition Department, University of Minnesota, 1334 Eckles Ave, Falcon Heights, MN 55108, USA; lars6588@umn.edu (R.L.); nels8935@umn.edu (C.N.); korcz005@umn.edu (R.K.); will2315@umn.edu (H.W.); eric2472@umn.edu (J.E.); 2Clinical and Translational Science Institute, University of Minnesota, 717 Delaware Street SE, Minneapolis, MN 55414, USA; wangx890@umn.edu

**Keywords:** gum acacia, gum arabic, satiety response, food intake regulation, dietary fiber, blood glucose, postprandial period, randomized controlled trial, crossover trials, healthy subjects

## Abstract

Acacia gum (AG) is a non-viscous soluble fiber that is easily incorporated into beverages and foods. To determine its physiological effects in healthy human subjects, we fed 0, 20, and 40 g of acacia gum in orange juice along with a bagel and cream cheese after a 12 h fast and compared satiety, glycemic response, gastrointestinal tolerance, and food intake among treatments. Subjects (*n* = 48) reported less hunger and greater fullness at 15 min (*p* = 0.019 and 0.003, respectively) and 240 min (*p* = 0.036 and 0.05, respectively) after breakfast with the 40 g fiber treatment. They also reported being more satisfied at 15 min (*p* = 0.011) and less hungry with the 40 g fiber treatment at 30 min (*p* = 0.012). Subjects reported more bloating, flatulence, and GI rumbling on the 40 g fiber treatment compared to control, although values for GI tolerance were all low with AG treatment. No significant differences were found in area under the curve (AUC) or change from baseline for blood glucose response, although actual blood glucose with 20 g fiber at 30 min was significantly less than control. Individuals varied greatly in their postprandial glucose response to all treatments. AG improves satiety response and may lower peak glucose response at certain timepoints, and it is well tolerated in healthy human subjects. AG can be added to beverages and foods in doses that can help meet fiber recommendations.

## 1. Introduction

Intake of dietary fiber is supported in dietary guidance for its role in reducing blood glucose and assisting in weight control. Fiber’s ability to lower blood glucose response plays a role in both prevention and management of Type 2 diabetes (T2D). One large multi-ethnic study followed 75,000 people for 14 years and found that those who consumed 15g/d or more of dietary fiber were significantly less likely to develop T2D [1]. Eating fiber may help prevent physiological signs of T2D and promote overall health in those with and without T2D [2,3,4,5].

As of 2016, nearly three-fourths (71.6%) of American adults are either overweight or obese [6]. Increased satiety, or the feeling of being satisfied after a meal, can prolong the time between meals and decrease the caloric intake at future meals [7,8]. Increasing dietary fiber intake may assist in obesity prevention and treatment due to its role in inducing satiety when consumed. Increased dietary fiber intake also correlates with lower body weight, body fat, and prevention of weight gain [7,8,9,10,11,12,13].

Dietary fibers have different effects on satiety and blood glucose based on dosage and physical properties, although few reliable patterns have emerged in research [7,14,15]. Decreased hunger, increased satiety, and decreased body weight have been demonstrated in studies using insoluble and soluble, viscous, and non-viscous fibers [7,8,16]. A large cohort study (*n* = 34,988) of older women without diabetes found a significant inverse effect of total dietary fiber (RR = 0.78, *p* = 0.005 comparing intakes of >23.6 and <15.3 g of total fiber per day) and insoluble fiber (RR = 0.75, *p* = 0.0012) consumption on diabetes risk, whereas soluble fiber showed no significant effect on diabetes risk (*p* = 0.23) until age- and energy-adjusted results were compiled (RR = 0.78, *p* = 0.0046) [17]. A 2017 systematic review of 12 randomized controlled trials examining effects of isolated soluble fibers (viscous and non-viscous, and fermentable and non-fermentable) on weight loss found that soluble fiber decreased body weight, body fat, fasting blood glucose, and fasting insulin significantly compared to the placebo [18]. No clear relationship has emerged between solubility, viscosity, molecular weight, or fermentability and strength of effects on appetite- or blood glucose-related outcomes [19]. Thus, fibers must be evaluated individually to determine the degree of their satiating properties and their impact on glycemia.

Acacia gum (AG), also known as gum arabic or gum acacia, is a soluble, non-viscous fiber made from the dried, powdered tree sap of *Acacia* tree species commonly found in Africa [20]. AG is often used as a stabilizer, thickening agent, and emulsifier in the food industry [20], but its effects on blood glucose, satiety, food intake, and weight status in healthy humans lack clarity. Limited research has shown benefits of AG consumption for satiety [21] and weight control [22,23,24] as well as glucose absorption and postprandial blood glucose response [24,25], although studies on weight control in humans come exclusively from one lab group [22,23,24] and studies on blood glucose control used only subjects already diagnosed with T2D. Overall, human and animal trials show excellent intestinal tolerance and safety for relatively high levels of AG supplementation. Human trials of AG doses up to 40g/d for four weeks in healthy adults, 30g/d for 6 weeks in university students and 3 months in adults with type 2 diabetes, and 25g/d AG over 12 weeks in healthy adults and adults with diabetic nephropathy showed no adverse effects, nor did a trial of 5% dietary level in rats [22,26,27,28].

The United States Food and Drug Administration (FDA) has set new requirements that isolated fibers must have beneficial physiological effects in healthy human subjects, such as a reduction in postprandial glycemic response, reduced energy intake from food, or increased subjective satiety, to be classified as “dietary fibers” for the purposes of nutrition labeling of food [29]. The primary objective of this study was to assess the effect of supplementation with 20 g or 40 g of AG from Nexira on satiety, including gastrointestinal tolerance and ad libitum food intake, in healthy human adults with effects on blood glucose response measured as a secondary objective. Our hypothesis is that AG will show beneficial physiological outcomes for the stated measurement areas and can qualify for classification as a dietary fiber by the FDA.

## 2. Materials & Methods

This study was reviewed by the Human Research Protection Program, IRB, University of Minnesota. The study was approved on 9 October 2018 and closed on 29 June 2020. The Assurance of Compliance number is FW A00000312.

All subjects gave their informed consent for inclusion before they participated in the study. The study was a randomized, double-blind, placebo-controlled, crossover trial (NCT03716479). Forty-eight subjects (24 men, 24 women) were recruited by flyers placed around the University of Minnesota campus. Sample size is based on power calculations (80% power with α = 0.05) calculated from the differences in visual analog scale (VAS) scores based on expected changes in satiety seen in previous studies in our lab with fiber treatments [30]. Subjects were screened and enrolled in the study with inclusion criteria: age 18–65 and a BMI between 18 and 29 kg/m^2^. Exclusion criteria: distaste for bagels and cream cheese, orange juice, or cheese pizza; current smoker; restrained eating habits (score of 11 or higher on a three-factor eating questionnaire [31]); recent weight change; any history of disease or significant past medical history; are vegetarian; do not normally eat breakfast or lunch; and are pregnant or lactating. All subjects were informed of the study participation requirements and they each signed an informed consent.

In the 24 h prior to each visit, subjects refrained from alcohol consumption and excessive exercise. Fasted subjects (12 h fasted) arrived at the University of Minnesota campus by 7:45 am for an 8 am breakfast. The initial visit and all subsequent visits included baseline anthropometrics prior to the test meal. Fasting blood glucose test must have been at or below 110 mg/dL prior to beginning the visit day. All visits were held in a quiet room in which subjects were able to read, use laptops, work quietly, or listen to music. Visits were scheduled at least 1 week apart. Subjects were randomized to receive each of three treatments, one per visit. Each treatment meal contains approximately 450 calories, 76 g carbohydrates, 2 g fiber, 13 g protein, and 6 g total fat, not including the AG fiber nutrient values:Treatment A (Control): 1 plain bagel, 1 oz (28.3 g) of cream cheese, and 8 fl. oz of orange juiceTreatment B: 1 plain bagel, 1 oz (28.3 g) of cream cheese, and 8 fl. oz of orange juice with 20 g of AG dissolvedTreatment C: 1 plain bagel, 1 oz (28.3 g) of cream cheese, and 8 fl. oz of orange juice with 40 g of AG dissolved

Computerized visual analog scales (VAS) were utilized to assess hunger, fullness, and desire to eat at baseline and periodically for four hours following the breakfast meal. Appetite sensations were rated by VAS at 15, 30, 45, 60, 90, 120, 180, and 240 min after baseline (Figure 1).

Subjects arrived at the test site on the morning after a 12 h fast, which was confirmed by a blood glucose test reading of <110 mg/dL. Subjects’ baseline blood glucose, satiety, and gastrointestinal status were assessed immediately prior to Time 0 (noted in the figure by a dot at Time 0). Immediately after assessment, subjects were given 15 min to consume the test breakfast meal (time for meal consumption is marked by diagonal lines). Satiety, gastrointestinal tolerance, palatability, and blood glucose were measured at regular intervals as shown in the figure. Subjects were given an ad libitum cheese pizza lunch at 240 min post-treatment. After 15 min, subjects were dismissed with instructions to keep a 24-h food log. This procedure was repeated at each of the three AG treatment administrations until all subjects had competed each of the three treatments. Treatments were administered one week apart for a total subject time of 6 weeks needed to complete the three treatments. Gastrointestinal tolerance was measured by subjective scale. GI tolerance questions assessed gas or bloating, nausea, flatulence, diarrhea or loose stools, constipation, gastrointestinal rumbling, and gastrointestinal cramping on a Likert-style rating scale of “none”, “mild”, “moderate”, “quite a lot”, “severe”, “very severe”, or “unbearable”. Gastrointestinal symptom surveys were completed at baseline, 60, 120, 180, and 240 min, 12 h and 24 h post-consumption (Figure 1). Treatment palatability ratings for visual appeal, smell, taste, aftertaste, and overall pleasantness were also assessed at 30 min using a sliding Likert-style rating scale of “good” to “bad”.

Blood glucose readings were collected at baseline, 30, 60, 120, 180, and 240 min by trained research assistants using the Bayer Contour Next EZ glucometer and sterile techniques (Figure 1). The first drop of blood was wiped away and the second drop of blood was collected for measurement.

Subjects were given an ad libitum cheese pizza lunch at 240 min post-treatment to assess prospective food intake and impact of AG on hunger/satiety. Subjects were instructed to “eat until comfortably full”. After 15 min, the remaining pizza was weighed and energy intakes were calculated.

Subjects were instructed to keep a detailed food record for 24 h following the study visit. Food records were analyzed with the Nutrition Data System for Research program (NDSR v.2018) for determination of total calories and other nutrients consumed. Nutrient values for foods reported as consumed that were missing from the NDSR database were extrapolated via finding the closest nutritional comparison item contained within the database.

Descriptive statistics were calculated and presented by treatment. Area under the curve (AUC) for VAS, ad libitum intake, gastrointestinal symptom score, and glucose was calculated using the trapezoidal rule. Only subjects with complete data were included in calculation of AUC. Satiety AUC, satiety at each time point, palatability, GI tolerance AUC, ad libitum calories, glucose AUC, baseline glucose, and change in glucose from time 0 to each time point were compared across the 3 treatments using mixed-effects models. Models included fixed effects of sequence, period, and treatment and a random effect of subject (nested within sequence) to account for the within-subject correlation among repeated measurements. The interaction between treatment and period was tested and dropped from the model as it was not significant. If the overall F test for the effect of treatment was significant, pairwise comparisons were conducted to examine which treatment is different from which and Tukey’s method was used to adjust for multiple comparisons. To examine whether males and females respond to the 3 treatments differently, treatment by gender interaction was tested for blood glucose, satiety VAS, and 24-hr food intake. Analyses were performed in SAS 9.3 (SAS Institute, Cary NC). Two-sided tests with *p*-value less than 0.05 were considered statistically significant.

## 3. Results

Statistical analysis included 48 participants: 24 men and 24 women. Subjects with incomplete data were not included in the final analysis for the area in which data were missing. Mean age for males was 27.6 ± 6.6 years and for females was 22.4 ± 2.0 years. Subjects had a mean BMI of 24.3 ± 2.0 for males and 23.5 ± 2.3 for females at baseline.

Even high doses of gum acacia were well tolerated (Table 1). Significantly more bloating, flatulence, and GI rumblings were reported for the 40 g fiber dose (treatment C) compared to the control (treatment A), although the overall values reported were relatively low. Other measures of GI tolerance were not altered by the gum acacia treatments. Palatability assessments between treatments were not significantly different (Table 1).

No significant difference (***p*** > 0.05) was found in blood glucose AUC or in change in blood glucose from baseline to 30, 60, 120, 180, and 240 min in any treatment. While actual blood glucose differed at 30 min overall (*p* = 0.011), the change in blood glucose from baseline to 30 min was not statistically significant (*p* = 0.16) (Table 2). Treatments B and C were indistinguishable at 30 min (*p* = 0.84), while B was significantly improved from the control group at 30 min (control = 144.2 mg/dL, treatment B = 136.7mg/dL, *p* = 0.013). Treatment C trended towards difference from control (*p* = 0.055). There was no significant interaction between treatment and gender. Glucose variability was found across all treatments (Figure 2). The widest variation among participants occurred between 30 and 60 min.

At time point 15 min, there was an overall significant difference in hunger (*p* = 0.025), satisfaction (*p* = 0.014), and fullness (*p* = 0.004) (Table 3). At 15 min post-breakfast consumption, subjects were less hungry with the 40 g fiber treatment compared to the control (*p* = 0.019). At the 15-min time point they were more satisfied with the 40 g fiber treatment compared to the control (*p* = 0.011). Additionally, at 15 min, the 40 g fiber treatment was associated with higher levels of fullness than the control (*p* = 0.003). At 30 min after breakfast, there was a statistically significant difference in hunger overall (*p* = 0.016). Subjects were less hungry after the 40 g fiber diet, compared to control at 30 min (*p* = 0.012). At the 240 min post-breakfast time point, subjects reported less hunger on the 40 g fiber treatment compared to the control (*p* = 0.036). They also reported being fuller at the 4 h time point with the 40 g fiber treatment compared to control (*p* = 0.05). Women in the 40 g fiber treatment reported more fullness AUC compared to the 20 g treatment (overall *p* = 0.048, A vs. B *p* = 0.033). Men in either the 20 g or 40 g group reported greater satisfaction at 15 min post-treatment than men in the control group (A vs. B *p* = 0.043; A vs. C *p* = 0.041). Women in the 40 g treatment group reported less hunger at 60 min than women in the control group (*p* = 0.012) or men in the 40 g treatment group (*p* = 0.011) (Table 4).

There were no significant differences in food intake at the pizza lunch among the three meals (Table 5). Calorie intake for control was 756 ± 274 kcal, 20 g fiber was 741 ± 230 kcal, and 40 g fiber was 737 ± 245 kcal (*p* value = 0.75).

Subjects also recorded their food intake for 24 h after the fiber treatments (Table 5). Calorie intakes trended downward, with calorie intakes being 1790 ± 749 kcal on control, 1625 ± 642 kcal on 20 g fiber, and 1590 ± 691 kcal on 40 g fiber treatment (*p* value = 0.12). Gender differences in this response were insignificant (*p* value = 0.16).

## 4. Discussion

The findings of this study indicate that gum acacia, a soluble fiber, has statistically significant positive effects on satiety measures at 15, 30, and 240 min after consumption. No significant effects on blood glucose AUC or ad libitum energy intake were seen, although 24-h food intake after the treatment showed a dose–response trend for lower self-reported food intake with increasing AG dose. We also found that AG added to orange juice at higher doses does not alter the acceptability of the orange juice. Additionally, large doses of AG had only minimal effects on subjects’ gastrointestinal tolerance, an effect mimicked in previous literature on AG tolerance [22,26,27,28].

Our study did not find significant effects of AG ingestion on blood glucose measures and suggests that AG does not produce a predictable glycemic response in healthy humans, which is inconsistent with previous literature [32]. A crossover study by Sharma et al. of similar design to this trial fed 0 or 20 g of AG to twelve healthy male subjects and showed a significant reduction (*p* < 0.05) in AUC blood glucose levels in subjects fed the AG compared to the control. Only acute effects of AG on the glycemic response were measured in both our study and the Sharma study. The lack of a significant reduction in mean glucose AUC after both doses of fiber in our trial could be due to interindividual glucose variability following consumption of AG; such variability in individual responses makes determining statistical power in blood glucose studies challenging, particularly in a healthy population that maintains a normal BG response. In this study, those with baseline blood glucose measurements 100–110 mg/dL maintained proportionally higher blood glucose data points throughout the study, signaling another area of interindividual variability difficult to account for in such studies. Limitations to this study include 1–5-min precision errors in blood glucose measurement timings; blood glucose was measured in a large group setting without a 1:1 ratio of participant to data collector. The young average age of subjects, the inability of researchers to control subject consumption and exercise prior to each trial, and the acute nature of the study are other limitations to consider.

While the type and viscosity of fiber can impact blood glucose levels after consumption, variation in the amount of fiber consumed cannot consistently predict the resulting reduction in postprandial glycemic response. The dose and the physical state of a food’s matrix may lead to variation in postprandial glucose and insulin responses [33,34]. Although there is a lack of conclusive research to demonstrate its physiological benefits long-term in healthy humans, AG consumption has shown glycemic benefits in people with T2D given 30 g supplement over a 3-month treatment period [24]. Use of AG in treatment of T2D is consistent with that of other low-GI foods; however, there is a vast amount of room for research to be done on soluble fibers including AG and their potential for management of those with T2D. The exact combination of fiber characteristics that will result in beneficial glycemic responses is largely unknown, thus requiring individual investigation for each fiber [35,36,37]. AG continues to be a minimally investigated fiber for its role in management of glycemic response.

Only one other study specifically examining the effects of AG on satiety was found at the time of this writing. Calame et al. found that supplementation of AG doses ranging from 5 to 40g resulted in a significant reduction in caloric intake compared to a negative control at an ad libitum meal three hours post-consumption of the AG. Additionally, a significant difference in subjective assessment of hunger and satiety was observed between all doses of AG and positive control. The largest dose (40g) resulted in the greatest difference in subjective satiety measures when compared to the negative control. However, there was no significant correlation between ad libitum intake and subjective satiety [21]. Clearly, more research is needed to properly assess the extent of AG’s influence on satiety and determine if any significant relationship exists between AG consumption and overall energy intake. This study is only the second known study to assess this question, lending it significance in the field of satiety and appetite research on dietary fibers.

Due to the variation of AG sample composition as a result of environmental factors [20], commercial standardization of samples is necessary before reliable experiments can be conducted and compared. As each commercial brand likely has a slightly different chemical composition, it is probable that effects of AG on ad libitum energy intake and subjective satiety are product-specific. Thus, the results of this study may not be directly comparable to those of the Calame study as different commercial fibers were utilized. While more research on AG effects on human physiology is needed, comparison of results may be challenging due to this factor.

Overall, AG has positive effects on body weight and adiposity in animals [38,39,40,41] and humans independently of subjective satiety, although only one lab is known at this time to study the effects of AG on body weight and adiposity in humans. Babiker and colleagues measured the effects of 30 g/d AG supplementation on various adiposity-related biomarkers in humans and found that AG supplementation led to statistically significant reductions in body mass index (BMI), body fat percentage, hip circumference, lipid accumulation product, and visceral adiposity [22,23,24]. Most astonishingly, the group saw a 23.7% (*p* = 0.022) decrease in visceral adiposity index (VAI) of their AG consuming group compared to the control over the course of the study [22]. Visceral fat accumulation measured by VAI is linked to impaired glucose and lipid metabolism, insulin resistance, and hypertension [22]. Thus, AG’s positive effect on satiety may not be the only beneficial mechanism by which AG could combat the rising rate of obesity.

## 5. Conclusions

This study indicates that gum acacia has physiological benefits in healthy human subjects, including improvements in satiety after consumption of 40 g of gum acacia per day with little effect on gastrointestinal comfort and high consumer acceptability. These results are consistent with existing research on AG and add to the body of evidence that AG provides physiological benefits as a functional fiber in human nutrition.

Fiber intakes in the United States remain at roughly half the recommended intake levels, and fiber remains a nutrient of concern in the Dietary Guidelines for Americans 2015–2020 [42]. Enrichment or fortification of fiber in food and beverage products could improve intakes of dietary fiber in America and have positive effects on health outcomes including obesity. The ability to add AG to beverages with high consumer acceptability and minimal gastrointestinal effects even at high doses may enable consumers to more easily meet their dietary fiber needs and demonstrates an advantage over other soluble fibers. Due to its high gastrointestinal tolerance, soluble nature, and effects on satiety, AG may be a practical fiber to help bridge the fiber gap in U.S. diets and have a positive effect on satiety.

## Figures and Tables

**Figure 1 nutrients-13-00618-f001:**
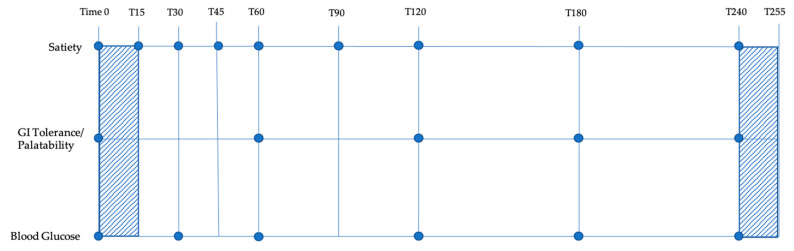
Data collection timeline on day of AG supplementation.

**Figure 2 nutrients-13-00618-f002:**
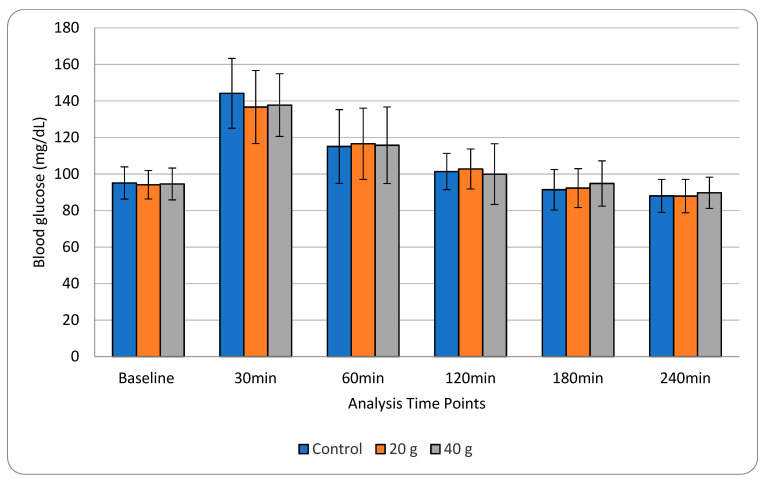
Mean blood glucose after breakfast treatment with 0, 20, and 40g of acacia gum. Error bars represent one standard deviation. Mean blood glucose differed significantly at 30 min (*p* = 0.011), although change in blood glucose from baseline to 30 min did not reach significance (*p* = 0.16). Treatment group B was significantly improved from the control group at 30 min (control = 144.2 mg/dL, treatment B = 136.7 mg/dL, *p* = 0.013). Treatment C trended towards difference from the control group at 30 min (*p* = 0.055).

**Table 1 nutrients-13-00618-t001:** Gastrointestinal tolerance and palatability.

	Treatment A			Treatment B			Treatment C			*p* Value			
	N	Mean	SD	N	Mean	SD	N	Mean	SD	Overall	A vs. B	A vs. C	B vs. C
Bloating AUC	47	5.4	7.9	48	6.9	10.8	46	12.3	15.0	**0.0008**	0.71	**0.0009**	**0.01**
Nausea AUC	47	0.5	1.9	48	0.6	2.9	46	1.9	6.2	0.11			
Flatulence AUC	47	4.0	7.4	48	7.8	11.1	46	11.5	15.5	**0.002**	0.18	**0.001**	0.14
Diarrhea/loose stools AUC	47	0.5	1.8	48	1.2	4.1	46	2.1	6.6	0.22			
Constipation AUC	47	0.9	3.7	48	1.4	5.5	46	3.2	7.2	**0.054**	0.86	**0.056**	0.17
GI rumbling AUC	47	6.4	8.8	48	7.4	10.1	46	11.4	14.3	**0.011**	0.81	**0.012**	**0.056**
GI cramping AUC	47	2.0	5.7	48	1.6	4.8	46	4.1	10.1	0.11			
Visual Appeal	44	0.21	0.19	46	0.28	0.21	43	0.28	0.23	**0.048**	0.086	0.078	1
Smell	44	0.25	0.17	46	0.28	0.18	44	0.28	0.18	0.39			
Taste	44	0.20	0.15	46	0.22	0.17	44	0.23	0.19	0.37			
Aftertaste	44	0.63	0.26	46	0.59	0.26	44	0.58	0.26	0.49			
Overall Palatability	44	0.22	0.16	46	0.24	0.20	44	0.28	0.22	0.11			

For gastrointestinal tolerance, *n* = 47 due to error in collection for one subject’s data. For palatability, *n* = 44 due to multiple errors in computer-based data collection where responses were not recorded. While visual appeal had overall significance, no significant differences emerged in the between-group breakout analysis. Bolded *p* values are considered significant (*p* < 0.05). Overall *p* value scores were calculated using the F test, with between-group comparisons calculated using Tukey’s method to adjust for multiple comparisons.

**Table 2 nutrients-13-00618-t002:** Blood glucose statistical analysis of three treatments across all time points.

	Treatment A (Control)			Treatment B			Treatment C			*p* Value
Glucose Outcomes	*N*	Mean	SD	*N*	Mean	SD	*N*	Mean	SD	-
AUC (time is in hour)	48	419.0	31.4	48	417.9	34.0	47	418.9	36.8	0.94
Baseline	48	95.1	8.8	48	92.6	14.5	48	94.5	8.6	0.49
30 min	48	144.2	19.1	48	136.7	20.0	48	137.9	17.0	**Overall: 0.011**
										**A vs. B: 0.013**
										A vs. C: 0.055
										B vs. C: 0.84
Baseline to 30 min change	48	49.1	18.3	48	44.1	23.5	48	43.5	15.9	0.16
Baseline to 60 min change	48	20.0	21.0	48	23.9	21.8	48	21.9	21.6	0.57
Baseline to 120 min change	48	6.3	12.0	48	10.1	16.1	48	5.2	18.1	0.28
Baseline to 180 min change	48	−3.7	10.3	48	−0.3	16.1	48	0.3	13.3	0.27
Baseline to 240 min change	48	−7.1	10.2	48	−4.7	16.7	47	−4.8	10.3	0.56

All blood glucose is measured in mg/dL using sterile techniques. Bolded *p* values are considered significant (*p* < 0.05). Significant values were only found for blood glucose levels at 30 min in Treatments B vs. the control, with levels in Treatment C vs. the control trending downwards but not reaching significance. Overall *p* value scores were calculated using the F test, with between-group comparisons calculated using Tukey’s method to adjust for multiple comparisons.

**Table 3 nutrients-13-00618-t003:** Satiety.

		Treatment A			Treatment B			Treatment C			*p* value			
	Outcome	*N*	Mean	SD	*N*	Mean	SD	*N*	Mean	SD	Overall	A vs B	A vs C	B vs C
Overall AUC	Hunger	42	1.9	0.55	38	1.85	0.55	43	1.75	0.54	0.22			
	Satisfaction	43	1.87	0.51	39	1.79	0.59	44	1.91	0.58	0.45			
	Fullness	43	1.82	0.54	39	1.79	0.61	44	1.96	0.68	0.15			
	Volume	43	2.2	0.6	39	2.26	0.68	44	2.12	0.72	0.37			
Baseline	Hunger	48	0.69	0.23	48	0.63	0.25	48	0.67	0.22	0.31			
	Satisfaction	48	0.22	0.22	48	0.22	0.22	48	0.19	0.16	0.58			
	Fullness	48	0.18	0.21	48	0.22	0.24	48	0.17	0.18	0.4			
	Volume	48	0.75	0.2	48	0.72	0.2	48	0.75	0.16	0.58			
15 minutes	Hunger	48	0.31	0.26	48	0.26	0.24	48	0.2	0.19	**0.025**	0.3	**0.019**	0.42
	Satisfaction	48	0.6	0.24	48	0.63	0.25	48	0.71	0.21	**0.014**	0.59	**0.011**	0.13
	Fullness	48	0.6	0.25	48	0.68	0.24	48	0.71	0.23	**0.004**	0.051	**0.003**	0.59
	Volume	48	0.41	0.26	48	0.37	0.26	48	0.36	0.26	0.33			
30 minutes	Hunger	46	0.3	0.24	47	0.24	0.2	46	0.2	0.19	**0.016**	0.2	**0.012**	0.45
	Satisfaction	47	0.66	0.21	48	0.66	0.2	47	0.69	0.22	0.59			
	Fullness	47	0.68	0.21	48	0.69	0.2	47	0.71	0.22	0.59			
	Volume	47	0.37	0.26	48	0.36	0.26	47	0.32	0.26	0.41			
45 minutes	Hunger	45	0.29	0.19	44	0.25	0.19	46	0.25	0.22	0.29			
	Satisfaction	46	0.66	0.18	45	0.66	0.2	46	0.65	0.23	0.98			
	Fullness	46	0.65	0.2	45	0.66	0.22	46	0.68	0.24	0.62			
	Volume	46	0.39	0.24	45	0.37	0.24	46	0.37	0.27	0.74			
60 minutes	Hunger	47	0.34	0.21	47	0.3	0.22	46	0.27	0.22	0.12			
	Satisfaction	48	0.6	0.19	48	0.6	0.21	47	0.62	0.24	0.78			
	Fullness	48	0.6	0.21	48	0.61	0.21	47	0.64	0.24	0.44			
	Volume	48	0.42	0.24	48	0.42	0.24	47	0.36	0.26	0.13			
90 minutes	Hunger	47	0.38	0.2	45	0.34	0.22	46	0.31	0.21	0.11			
	Satisfaction	47	0.57	0.2	45	0.55	0.2	47	0.58	0.23	0.83			
	Fullness	47	0.55	0.22	45	0.53	0.23	47	0.62	0.21	0.071			
	Volume	47	0.47	0.22	45	0.47	0.22	47	0.41	0.25	0.19			
120 minutes	Hunger	47	0.44	0.23	47	0.43	0.22	47	0.41	0.23	0.73			
	Satisfaction	47	0.47	0.19	47	0.47	0.21	47	0.51	0.23	0.51			
	Fullness	47	0.46	0.22	47	0.47	0.22	47	0.52	0.24	0.24			
	Volume	47	0.54	0.21	47	0.52	0.22	47	0.48	0.23	0.32			
180 minutes	Hunger	48	0.58	0.22	46	0.59	0.21	48	0.56	0.2	0.7			
	Satisfaction	48	0.36	0.21	46	0.33	0.2	48	0.36	0.18	0.49			
	Fullness	48	0.35	0.21	46	0.32	0.21	48	0.35	0.22	0.45			
	Volume	48	0.68	0.18	46	0.7	0.17	48	0.66	0.2	0.46			
240 minutes	Hunger	48	0.79	0.13	48	0.78	0.14	48	0.73	0.16	**0.035**	0.84	**0.036**	0.13
	Satisfaction	48	0.17	0.12	48	0.18	0.12	48	0.2	0.13	0.34			
	Fullness	48	0.17	0.17	48	0.18	0.16	48	0.24	0.21	**0.046**	0.9	**0.05**	0.13
	Volume	48	0.82	0.14	48	0.8	0.13	48	0.79	0.15	0.26			

Bolded *p* values are considered significant (*p* < 0.05).

**Table 4 nutrients-13-00618-t004:** Satiety results by gender.

		Treatment A						Treatment B						Treatment C						
		Female			Male			Female			Male			Female			Male			*p* Value
	Outcome	*N*	Mean	SD	*N*	Mean	SD	*N*	Mean	SD	*N*	Mean	SD	*N*	Mean	SD	*N*	Mean	SD	
Overall AUC	Hunger	21	1.88	0.5	21	1.92	0.6	20	1.8	0.57	18	1.92	0.54	22	1.57	0.51	21	1.93	0.52	0.14
	Satisfaction	22	1.88	0.4	21	1.86	0.61	21	1.72	0.6	18	1.86	0.58	23	2.06	0.55	21	1.75	0.58	**0.033**
	Fullness	22	1.84	0.46	21	1.79	0.62	21	1.73	0.64	18	1.86	0.59	23	2.12	0.68	21	1.8	0.66	**0.048**
	Volume	22	2.16	0.43	21	2.24	0.75	21	2.25	0.62	18	2.27	0.77	23	1.93	0.6	21	2.32	0.79	**0.039**
Baseline	Hunger	24	0.71	0.19	24	0.68	0.27	24	0.62	0.28	24	0.65	0.23	24	0.65	0.24	24	0.69	0.2	0.79
	Satisfaction	24	0.21	0.23	24	0.23	0.22	24	0.19	0.21	24	0.25	0.22	24	0.19	0.18	24	0.18	0.14	0.64
	Fullness	24	0.18	0.24	24	0.18	0.19	24	0.18	0.22	24	0.27	0.26	24	0.2	0.22	24	0.15	0.14	0.23
	Volume	24	0.76	0.2	24	0.74	0.21	24	0.73	0.2	24	0.72	0.2	24	0.72	0.17	24	0.78	0.16	0.41
15 minutes	Hunger	24	0.26	0.21	24	0.35	0.3	24	0.23	0.23	24	0.28	0.25	24	0.18	0.2	24	0.23	0.19	0.69
	Satisfaction	24	0.66	0.19	24	0.54	0.28	24	0.58	0.28	24	0.69	0.2	24	0.72	0.2	24	0.7	0.22	**0.008**
	Fullness	24	0.67	0.19	24	0.52	0.29	24	0.68	0.24	24	0.67	0.24	24	0.74	0.21	24	0.69	0.24	0.084
	Volume	24	0.37	0.21	24	0.45	0.3	24	0.37	0.25	24	0.37	0.27	24	0.32	0.23	24	0.39	0.28	0.55
30 minutes	Hunger	22	0.27	0.24	24	0.33	0.24	23	0.23	0.22	24	0.26	0.18	22	0.13	0.13	24	0.27	0.22	0.16
	Satisfaction	23	0.71	0.17	24	0.61	0.25	24	0.68	0.2	24	0.64	0.21	23	0.73	0.2	24	0.66	0.23	0.54
	Fullness	23	0.73	0.16	24	0.62	0.23	24	0.71	0.2	24	0.66	0.2	23	0.77	0.19	24	0.65	0.24	0.24
	Volume	23	0.32	0.24	24	0.43	0.28	24	0.35	0.25	24	0.38	0.27	23	0.29	0.24	24	0.36	0.27	0.47
45 minutes	Hunger	23	0.22	0.13	22	0.36	0.22	23	0.2	0.2	21	0.31	0.18	23	0.16	0.15	23	0.34	0.25	0.53
	Satisfaction	24	0.69	0.16	22	0.63	0.21	24	0.68	0.22	21	0.64	0.18	23	0.7	0.21	23	0.6	0.24	0.57
	Fullness	24	0.67	0.19	22	0.63	0.21	24	0.69	0.22	21	0.61	0.22	23	0.75	0.2	23	0.61	0.27	0.25
	Volume	24	0.38	0.22	22	0.41	0.27	24	0.35	0.25	21	0.39	0.24	23	0.33	0.26	23	0.42	0.29	0.68
60 minutes	Hunger	23	0.31	0.2	24	0.37	0.23	23	0.23	0.21	24	0.36	0.21	22	0.16	0.14	24	0.38	0.24	**0.027**
	Satisfaction	24	0.63	0.17	24	0.58	0.22	24	0.63	0.22	24	0.58	0.2	23	0.69	0.22	24	0.56	0.25	0.29
	Fullness	24	0.66	0.17	24	0.55	0.23	24	0.66	0.2	24	0.55	0.21	23	0.72	0.21	24	0.57	0.24	0.74
	Volume	24	0.39	0.22	24	0.44	0.27	24	0.37	0.24	24	0.47	0.24	23	0.26	0.2	24	0.44	0.28	0.12
90 minutes	Hunger	24	0.31	0.17	23	0.44	0.21	22	0.28	0.21	23	0.41	0.22	23	0.23	0.18	23	0.39	0.2	0.62
	Satisfaction	24	0.6	0.18	23	0.53	0.22	22	0.57	0.22	23	0.52	0.18	24	0.6	0.24	23	0.55	0.22	0.96
	Fullness	24	0.59	0.24	23	0.51	0.21	22	0.57	0.25	23	0.5	0.21	24	0.67	0.2	23	0.56	0.21	0.81
	Volume	24	0.43	0.2	23	0.52	0.24	22	0.44	0.23	23	0.5	0.22	24	0.35	0.24	23	0.47	0.25	0.57
120 minutes	Hunger	23	0.41	0.23	24	0.47	0.23	24	0.39	0.21	23	0.47	0.22	24	0.33	0.23	23	0.49	0.19	0.37
	Satisfaction	23	0.49	0.18	24	0.45	0.21	24	0.49	0.22	23	0.45	0.21	24	0.6	0.21	23	0.41	0.2	0.059
	Fullness	23	0.47	0.24	24	0.44	0.21	24	0.48	0.24	23	0.47	0.2	24	0.59	0.25	23	0.45	0.2	0.25
	Volume	23	0.52	0.18	24	0.56	0.24	24	0.49	0.23	23	0.55	0.21	24	0.4	0.22	23	0.57	0.22	0.19
180 minutes	Hunger	24	0.61	0.2	24	0.55	0.24	23	0.59	0.24	23	0.6	0.19	24	0.57	0.21	24	0.56	0.19	0.53
	Satisfaction	24	0.35	0.19	24	0.37	0.24	23	0.32	0.21	23	0.33	0.18	24	0.38	0.18	24	0.33	0.18	0.48
	Fullness	24	0.35	0.2	24	0.36	0.23	23	0.31	0.24	23	0.33	0.17	24	0.37	0.24	24	0.32	0.19	0.49
	Volume	24	0.69	0.11	24	0.67	0.22	23	0.71	0.13	23	0.68	0.2	24	0.63	0.2	24	0.69	0.2	0.2
240 minutes	Hunger	24	0.8	0.13	24	0.78	0.13	24	0.82	0.09	24	0.74	0.17	24	0.74	0.17	24	0.72	0.15	0.28
	Satisfaction	24	0.15	0.11	24	0.19	0.13	24	0.15	0.1	24	0.21	0.13	24	0.19	0.13	24	0.21	0.14	0.48
	Fullness	24	0.13	0.13	24	0.22	0.2	24	0.12	0.1	24	0.24	0.18	24	0.22	0.21	24	0.26	0.21	0.39
	Volume	24	0.81	0.15	24	0.82	0.14	24	0.81	0.1	24	0.78	0.15	24	0.79	0.15	24	0.78	0.15	0.6

Bolded *p* values are considered significant (*p* < 0.05). Women in the 40 g fiber treatment reported more fullness area under the curve (AUC) compared to the 20 g treatment (A vs. B *p* = 0.033). While overall *p* values were significant for satisfaction and volume AUC, no other between-group analyses emerged as significant. Men in either the 20 g or 40 g group reported greater satisfaction at 15 min post-treatment than men in the control group (A vs. B *p* = 0.043; A vs. C *p* = 0.041). At 60 min, women in the 40 g treatment group reported less hunger than women in the control group (*p* = 0.012) or men in the 40 g treatment group (*p* = 0.011).

**Table 5 nutrients-13-00618-t005:** Ad libitum pizza calories consumed and total calorie intake from 24-hr recall by treatment group.

	Treatment A			Treatment B			Treatment C			
Outcome	*N*	Mean	SD	*N*	Mean	SD	*N*	Mean	SD	*p* Value
AL Pizza calories	48	756.5	274.1	47	741.7	229.7	48	737.0	243.5	0.75
24-hr Total calories	48	1790	749	48	1625	642	48	1590	691	0.12
Female: Total calories	24	1721	762	24	1516	612	24	1676	732	0.16
Male: Total calories	24	1859	747	24	1734	666	24	1505	651	0.16

Treatment by gender interaction *p* value is 0.16, indicating no gender differences.

## Data Availability

The data presented in this study are available on request from the corresponding author within the terms of the contract that funded the study. Funding details are listed above.

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
