# Peer review of "Acacia Gum Is Well Tolerated While Increasing Satiety and Lowering Peak Blood Glucose Response in Healthy Human Subjects"

_nutrients, 2021, doi:10.3390/nu13020618_

Round 1

Reviewer 1 Report

This manuscript clearly presented the findings of the study. 

There are a couple minor grammatical issues. For instance, in the "Discussion" section, paragraph 2, consider changing "suggest" to "suggests." 

Other than that, a graphical representation of the experiment time line would be helpful.  I am seeing these increasingly in the literature, and they help with speed and clarity. 

Author Response

Grammatical errors have been corrected.

A study timeline is included.

Reviewer 2 Report

Larson and colleagues present data from a small cross over trial testing the hypothesis that add on of a known amount of fibers from acacia gum could reduce glucose excursion and eventually promote satiety in healthy patients. 

Overall the study is well designed, results are presented clearly and the paper is easy to read. I have only some minor recommendations to improve the completeness and clarity of the work:

1- In the method section, there is no specification of the sample size calculation. Why the number 48 was chosen? Even if calculation has not been performed at this stage, please cite a previous work with a similar setting that obtained significant results with such a comparable number of patients. Also, this should imply that one outcome is more relevant compared to another, since the study should be designed to test one primary outcomes while the others should be treated as exploratory. Please provide more details on which was the primary outcome of interest (glucose AUC or satiety?).

2- Method section: it is stated that both F test and Tukey's test have been used. However, it is unclear which has been used for which of the variables. Please amend it directly in the captions of table 1 and 2 (when p values are described it should be stated from which test do they derive).

3- Figure 2,3 and 4 are not really helpful. Albeit it is well known there is a large inter- and intra-individual variability of glucose oscillations according to different meals, you might build a unique graph comparing 0, 20, 40 g of acacia gum by using the mean +- sem (or sd, better) of all subjects at any time point. This would allow to easiliy observe the eventual difference (if i understand correctly only one time point comparing 20 grams vs no gum).

4- Minor point: individual points should be displayed instead of histograms (or empty histograms containing also the individual points) in figure 1.

5- minor but mandatory: dietary composition of bagel and cheese should be clearly and precisely defined. This breakfast may be typical in US, Poland and among Yiddish, but it is not common elsewhere in the world (surely not in the Mediterranean Europe and likely not in Asia). Albeit this info might be obtained from other sources, it would be desirable to known the exact nutrient composition of the food provided in the trial. 

Author Response

1.  We powered the study with satiety as the primary endpoint.  We had no preliminary data on the effect size with AG on satiety, so we used sample sizes for past satiety studies we had done on soluble, non-viscous fibers. 

2.  We used different statistical methods on some of the secondary endpoints, which led to the unclear statistical treatment of the data that you described.  Qi Wang, our statistician, has helped revise this section for clarity.

3.  Me agree.  The Figures were used to show the wide variation in response to blood glucose.  They have been removed.

4.  The histograph has been changed as suggested.

5.  The bagel breakfast treatment goes across all treatments - which allows us to test the effect of the fiber on our primary and secondary endpoints.  We agree that few countries have a bagel for breakfast.  So the nutritional data on the breakfast is included, although since it is standardized across treatments.
